# Utility of Stool-Based Tests for Colorectal Cancer Detection: A Comprehensive Review

**DOI:** 10.3390/healthcare12161645

**Published:** 2024-08-18

**Authors:** Raquel Gómez-Molina, Miguel Suárez, Raquel Martínez, Marifina Chilet, Josep Miquel Bauça, Jorge Mateo

**Affiliations:** 1Department of Laboratory Medicine, Virgen de la Luz Hospital, 16002 Cuenca, Spain; 2Gastroenterology Department, Virgen de la Luz Hospital, 16002 Cuenca, Spain; 3Medical Analysis Expert Group, Institute of Technology, Universidad de Castilla-La Mancha, 16071 Cuenca, Spain; 4Medical Analysis Expert Group, Instituto de Investigación Sanitaria de Castilla-La Mancha (IDISCAM), 45071 Toledo, Spain; 5Department of Laboratory Medicine, Hospital Universitari Son Espases, 07120 Palma, Spain

**Keywords:** colorectal cancer, screening, cancer early detection, fecal occult blood test

## Abstract

Colorectal cancer (CRC) is a significant global health issue where early detection is crucial for improving treatment outcomes and survival rates. This comprehensive review assesses the utility of stool-based tests in CRC screening, including traditional fecal occult blood tests (FOBT), both chemical (gFOBT) and immunochemical techniques (FIT), as well as multitarget stool DNA (mt-sDNA) as a novel and promising biomarker. The advancements, limitations and the impact of false positives and negatives of these methods are examined. The review analyzed various studies on current screening methods, focusing on laboratory tests and biomarkers. Findings indicate that while FIT and mt-sDNA tests offer enhanced sensitivity and specificity over traditional guaiac-based FOBT, they also come with higher costs and potential for increased false positives. FIT shows better patient adherence due to its ease to use, but incorrect usage and interpretation of FOBT can lead to significant diagnostic errors. In conclusion, despite the improvements in FOBT methods like FIT in CRC detection, careful consideration of each method’s benefits and drawbacks is essential. Effective CRC screening programs should combine various methods tailored to specific population needs, aiming for early detection and reduced mortality rates.

## 1. Introduction

Colorectal cancer (CCR) is currently the fourth most prevalent malignant tumor, representing a significant cause of morbidity and mortality worldwide and a major public health issue in Westernized countries. It accounts for nearly 10% of all new cancer cases diagnosed in 2023 and ranks as the fifth most common cause of cancer-related deaths, with approximately 550,000 deaths annually worldwide [1]. In the United States of America (USA), CRC ranks as the third most prevalent cancer and the third leading cause of cancer-related deaths [2]. In Spain, it accounts for 15% of detected cancers, with over 25,000 new cases last year, and it is the second leading cause of cancer death (over 13,000 deaths annually) [3,4]. Similar figures are observed across Europe, where it presents a five-year average survival rate of 57% from diagnosis [5]. Incidence is slightly higher in men than in women, being the second most common cancer in males after prostate cancer and in females after breast cancer. When considering both sexes together, colorectal cancer is the most common cause of cancer death. Both the mortality and incidence of colorectal cancer are age-related, with approximately 90% of new cases diagnosed and 94% of deaths occurring in individuals over 50 years old [6,7].

Patients diagnosed at a localized stage have a much higher survival rate compared to those with metastasis at the time of diagnosis. Risk factors for CRC include a history of chronic diseases (such as inflammatory bowel disease (IBD)), sedentary lifestyle, age, obesity, unhealthy dietary habits, and substance use (tobacco and alcohol) [8]. Therefore, the increasing prevalence of CRC in industrialized countries can be attributed to an aging population, unhealthy dietary habits, and an increase in risk factors such as smoking, physical inactivity, and obesity [8,9]. Additionally, approximately 5% of individuals with colorectal cancer harbor inherited genetic mutations that can lead to familial cancer syndromes. The most common hereditary syndromes linked to colorectal cancer include Lynch syndrome (hereditary nonpolyposis colorectal cancer or HNPCC) and familial adenomatous polyposis (FAP), though other less common syndromes also increase the risk [8,10]. Lynch syndrome, while representing less than 5% of all colorectal cancer cases, significantly raises the risk of colorectal cancer compared to the general population (82% vs. 5%) [11,12].

Fecal Occult Blood Testing (FOBT) is a fundamental tool in the screening and early diagnosis of colorectal cancer and other gastrointestinal diseases. This non-invasive biomarker allows the identification of small amounts of blood that are not visible to the naked eye, which may indicate premalignant or malignant lesions in the gastrointestinal tract [13]. Although FOBT are primarily validated for CRC screening for secondary prevention, they are commonly misused in other clinical settings as a diagnostic test. This can lead to inappropriate clinical decisions and unnecessary follow-up procedures, highlighting the importance of proper interpretation guidelines and clinical contexts [14,15]. Additionally, multitarget stool DNA (mt-sDNA) is also being introduced as a screening technique for CRC due to its high sensitivity and potential to improve early detection rates. Adherence is crucial for the success of screening programs, as non-compliance with repeated tests can compromise their effectiveness in detecting early signs of colorectal cancer [16].

The aim of this review is to conduct a thorough analysis of current colorectal cancer screening strategies, evaluating their adherence and effectiveness. Additionally, it will examine diverse laboratory techniques employed in clinical settings throughout past, present, and future methodologies, comparing their sensitivity, specificity, and rates of false positives and negatives. Finally, this review will critically assess the inappropriate utilization of these techniques in hospitalized patients and/or unsuitable candidates for screening, emphasizing the risks associated with misinterpretation.

## 2. Current Review

In this review, we analyzed studies published in the scientific literature until 2024. We examined various CRC screening strategies, focusing on the underlying principles and differences among the tests. The review includes FOBT based on both chemical and immunochemical methods, as well as other biomarkers such as multitarget tests for genetic and epigenetic alterations in fecal DNA. Our goal is to provide a comprehensive understanding of the effectiveness and limitations of these screening methods.

### 2.1. Screening Strategies for Colorectal Cancer

Regular participation in screening programs has been shown to reduce CRC mortality, as it is a common and preventable malignancy. Clinical practice guidelines recommend routine CRC screening for average-risk individuals starting at age 45 [5,17]. However, in recent years, there has been an alarming increase in incidence among those under 50 years, at a rate of 2 to 4% annually in many countries, expected to become the leading cause of cancer death in individuals aged 20 to 49 in the US by 2030 [18]. In response, recommendations from the US Preventive Services Task Force (USPSTF) propose lowering the screening age to 40 [19]. In Europe, the recommended screening age has been extended up to 75 years for both men and women, though implementation varies across European Union (EU) countries [20]. For adults aged 76 and older, the balance between benefits and risks of colorectal cancer screening becomes less favorable, and screening varies based on patient health status (e.g., life expectancy, comorbid conditions, prior screening history, and individual preferences) [3,19]. Screening strategies for colon cancer vary significantly across healthcare systems, influenced by factors such as available resources, local medical guidelines, and patient preferences. These variations can impact the availability, frequency, and types of screening tests offered to the population [21,22].

The development of CRC begins with alterations in healthy colonic epithelium, initiating the formation of adenomatous polyps. These polyps can proliferate and grow, accumulating genetic and epigenetic mutations over time [23]. The adenoma-carcinoma sequence describes how some of these polyps with malignant characteristics can progress to invasive cancer, with the potential to metastasize. Importantly, not all adenomas develop invasive cancer [24]. In contrast, serrated polyps follow a different sequence and pose unique challenges in terms of diagnosis and clinical management [25]. Due to their uncontrolled growth, adenomatous polyps can invade surrounding tissues, particularly the intestinal wall, and eventually spread through the lymphatic and circulatory systems. Early identification and removal of these polyps aim to significantly reduce the risk of developing CRC and associated mortality [26,27]. Since symptoms of CRC are not reliable predictors of the disease in early stages, colonoscopy is considered the gold standard diagnostic test. However, it is an invasive, costly procedure with potential complications [2].

There are other multiple techniques available for CRC screening: flexible sigmoidoscopy (FS); FOBT; computed tomography colonoscopy (CTC); and capsule colonoscopy (CCE) [21,28,29]. One way to classify CRC screening tests is to divide them into one-step tests, such as colonoscopy, which are both diagnostic and therapeutic; and two-step tests, which require a colonoscopy if the initial result is positive to complete the screening process and include the rest of the techniques (FS, FOBT, CTC and CCE). The requirement for a subsequent colonoscopy after a positive result is a significant drawback of non-colonoscopy-based CRC screening tests. Individuals who refuse or cannot undergo colonoscopy or FIT, as well as those with incomplete colonoscopies, should resort to one of the additional two-step tests, such as FS, multitarget stool DNA test (mts-DNA), CTC, or CCE [14,30,31].

Screening strategies based on stool tests rely on patients completing the tests regularly and on time. Non-compliance with repeated screenings can compromise their effectiveness. The review by Murphy et al. shows a significant prevalence of completing two or more rounds of FOBT; the proportion of individuals who completed FOBT in two rounds ranged from 16.4% to 80.0% (median: 46.6%; IQR: 40.5–50.0%). However, the repeat rate of FOBT decreased significantly in successive screening rounds. Additionally, repeat FOBT appeared to be higher in mailed outreach programs (69.1–89.6%) compared to opportunistic screening (24.6–48.6%) [28]. Repeating FOBT in clinical practice is also complex, as it requires reevaluating patient eligibility, considering recommended intervals (annual vs. biennial), and identifying patients who should undergo screening in each round [26,32]. Furthermore, it is noteworthy that the meta-analysis conducted by Jodal et al. over a 15-year period revealed that screening with FOBT reduced colorectal cancer-related mortality but did not decrease the incidence of this type of cancer [33].

Additionally, adherence to screening is crucial for reducing mortality of CRC. In this regard, a disparity in adherence to FOBT screening has also been observed based on the socioeconomic level of the population, with lower adherence among lower socioeconomic classes [34]. To optimize the effectiveness of this strategy, it is essential to implement interventions that increase and maintain participation in screening, as well as interventions that help reduce the disparity in adherence between high and low socioeconomic levels. Most European countries [35], Canada [36], and Australia recommend biennial stool-based screening, while in USA and Asian countries, annual screening is recommended [37,38].

### 2.2. Stool-Based Tests

There are multiple screening tests available to detect CRC and adenomatous polyps. Stool-based detection of CRC is quite straightforward, cost-effective, and the least invasive screening method available [39]. FOBT, which detects hemoglobin enzymatically or immunologically, is the most widely used screening modality for CRC. These tests involve collecting and analyzing a stool sample to detect the presence of hidden blood, which could indicate underlying CRC [40]. The concept of analyzing stool samples for occult blood began in the early 1900s, and numerous studies since then have documented the efficacy of this test in detecting the presence of this type of cancer [14]. There are several types of fecal occult blood tests available, such as the guaiac-based Fecal Occult Blood Test (gFOBT), which is designed to detect the peroxidase enzyme activity, present in human red blood cells; and FIT, which is based on detecting the protein called globin, also present in blood erythrocytes [41,42] (Figure 1).

The fecal occult blood test is cost-effective, non-invasive, and straightforward, making it an accessible option for large populations [43,44]. However, there are limitations, such as the need for annual test repetition and lower sensitivity in detecting advanced adenomas compared to other techniques [45,46]. In this regard, specificity and sensitivity are crucial metrics in evaluating the performance of diagnostic tests. Sensitivity refers to the test’s ability to correctly identify those with the disease (true positive rate), while specificity measures the test’s ability to correctly identify those without the disease (true negative rate). A false positive occurs when the test erroneously suggests disease presence in a healthy individual, whereas a false negative happens when the test fails to detect the disease in an affected individual. These values are essential for interpreting test results accurately and understanding the clinical implications of test findings [47]. In this section, we will examine stool-based tests to detect hemoglobin in blood that could come from a lesion or DNA alterations suggestive of malignancy.

#### 2.2.1. Guaiac-Based Fecal Occult Blood Test (gFOBT)

The guaiac-based Fecal Occult Blood Test (gFOBT) detects hemoglobin through a chemical reaction that turns a paper impregnated with a guaiac reagent blue. It is a chromogenic and qualitative test that provides a positive or negative result. It detects occult blood through a peroxidase reaction where hydrogen peroxide catalyzes the oxidation of guaiac, producing a blue color if hemoglobin is present. The test is conducted using a card with guaiac-impregnated paper, a phenolic compound. When a stool sample is applied to the paper and a hydrogen peroxide developer solution is added, the hematin portion of the hemoglobin (Hb) in the blood catalyzes the release of oxygen, which in turn oxidizes the guaiac and results in a blue color change [32,48]. gFOBT does not detect Hb concentrations of less than approximately 600 µg Hb/g faeces; when gFOBT is rehydrated, the analytical sensitivity is higher, but more false-positive results are obtained [49,50]. Additionally, it requires three consecutive samples, making the sample collection process longer compared to other tests.

Although generally no special preparation is needed, certain guidelines recommend avoiding interference with the results [41,50]. There is no need for a restrictive diet, but it is advisable to eliminate red meat for three days to avoid the risk of meat residues appearing in the stool samples collected [51]. Vitamin C intake should also be restricted to less than 250 mg per day, as high doses can cause false-negative results. The positive predictive value of gFOBT may be altered if the patient is being treated with non-steroidal anti-inflammatory drugs (NSAIDs) such as ibuprofen, naproxen, and aspirin [52]. Collecting samples during a digital rectal exam, menstruation, or if the patient has hemorrhoids is not advisable due to the risk of false-positive results. Additionally, there are other factors that can affect the reading of the gFOBT, such as interobserver variability, reproducibility, temperature, the design of the gFOBT card, and lighting [49].

Studies have shown that screening with gFOBT reduces CRC mortality. In a meta-analysis conducted by Zhang et al., within a sub analysis involving 19 studies and 2,264,603 participants comparing gFOBT with no screening, a 14% reduction in CRC mortality was observed (RR, 0.86; 95% CI, 0.82–0.90) [53]. In randomized trials and observational studies using different FOBTs, the sensitivity of gFOBT for CRC detection ranged from 31% to 79%, and specificity ranged from 87% to 98% [53].

#### 2.2.2. Fecal Immunochemical Test (FIT)

The Fecal Immunochemical Test (FIT) is an immunoassay designed to detect human hemoglobin in stool samples. It works by using specific antibodies that bind to human hemoglobin, allowing for the detection of very low levels of fecal blood (40–300 µg/g) [54,55,56]. The test is performed by providing a stool sample in a specially designed container, without requiring any prior dietary or medication restrictions [57,58]. The results can be either qualitative or quantitative, offering a sensitive and non-invasive method for colorectal cancer screening. It is recommended that patients return the sample within 24 h of collection, as the sensitivity of FIT decreases with delayed return after sampling. The simplicity of sample collection has proven to be a significant factor in enhancing patient adherence to screening programs [59]. Unlike gFOBT, foods containing peroxidase activity do not cause false positives, and discontinuation of NSAIDs is generally unnecessary. However, the use of proton-pump inhibitors (PPIs) has been found to increase FIT positivity at the expense of false positive results [60]. FIT is typically performed annually in the USA and less frequently in other countries [20,32,36]. In a randomized trial in the Netherlands, screening with FIT (single sample) every three years, compared to annual or biennial screening, resulted in similar positivity rates and slightly higher participation in the second screening round [61].

Fecal immunoassays (FIT) have gained popularity due to their ability to detect human globin, reducing interference from external factors and enhancing specificity for lower gastrointestinal bleeding. Among the advantages of this type of test, its higher sensitivity stands out. The meta-analysis conducted by Lee et al. evaluating FIT established a sensitivity of 79% (95% CI 0.69–0.86) for one-time testing and a specificity of 94% (95% CI 0.92–0.95) [62]. However, sensitivity and specificity of FIT for detecting advanced adenoma are lower, estimated between 25–56% and 68–96%, respectively. Furthermore, although FIT has higher detection rates for CRC and advanced adenomas compared to gFOBT, it may be less sensitive for right-sided colon lesions. Additionally, a positive FIT has high specificity for lower gastrointestinal bleeding [59,60]. Regarding repeated testing, it has been shown to improve sensitivity, although it likely decreases specificity and positive predictive value (PPV) [41].

#### 2.2.3. Other Biomarkers: Multitarget Stool DNA Test (mt-sDNA)

In addition to fecal occult blood testing, there are other techniques used for the detection and monitoring of CRC. Among them are tests for genetic and epigenetic alterations in fecal DNA, which have been considered as a potential method for early detection of CRC. These tests analyze DNA released from tumor cells present in the patient’s stool. The presence of mutations in Kirsten rat sarcoma viral oncogene homolog (KRAS), and aberrant hypermethylation of promoter regions N-Myc Downstream Regulated Gene 4 (NDRG4) and Bone Morphogenic Protein 3 (BMP3) [42,43], are closely associated with the presence of CRC and advanced adenomas. For example, KRAS mutation is found in 30–40% of CRC tumors [63,64], so it makes fecal DNA test a promising tool for screening and early detection of this disease [19]. The multitarget stool DNA (mt-sDNA) test includes a panel of quantitative molecular assays for KRAS gene mutations and aberrant methylation of NDRG4 and BMP3 in addition to hemoglobin immunoassays (FIT) [65].

The mt-sDNA test uses a single random stool sample collected at home without requiring diet or medication changes. It detects 10 biomarkers associated with CRC and precancerous lesions, including altered DNA and hemoglobin [66]. The test combines these results with the beta-actin gene in an algorithm to produce a single qualitative “negative or positive” result. Logistic regression algorithms are employed to establish a specific threshold value [67]. It is important to note that there is no quantitative linear relationship between the threshold value and cancer progression. mt-sDNA tests were developed to offer improved sensitivity for detecting CRC and significant precancerous lesions compared to tests that rely solely on fecal hemoglobin. Unlike fecal hemoglobin tests, which have limitations in detecting neoplasms due to intermittent bleeding or lack of bleeding, mt-sDNA identifies DNA alterations, such as mutations and aberrant methylation, from cells shed by lesions and cancers at various stages. This enhances the test’s ability to detect these conditions. Additionally, mt-sDNA minimizes sampling errors by using a homogeneous stool sample and a standardized laboratory processing protocol, thereby increasing detection accuracy compared to FIT and gFOBT tests, which suffer from sampling errors due to their reliance on random sampling and smaller sample sizes [65,68] (Figure 2).

The availability of mt-DNA provides a highly sensitive and non-invasive option for colorectal cancer screening, particularly beneficial for patients who cannot or prefer not to undergo colonoscopies due to personal, cultural, or geographic limitations [65,69]. This allows for early identification of colorectal neoplasms without requiring strict annual follow-up regimens. Compared to FIT, mt-sDNA has a sensitivity of 93% for colorectal cancer stages I-III, and sensitivity increases with lesion size and the grade of adenomas and sessile serrated polyps [70]. mt-sDNA sensitivity is equivalent for colorectal cancer in both the proximal and distal colon, unlike FIT, which is more sensitive for distal lesions. This is significant for reducing the incidence of proximal CRC through non-invasive screening [41,49]. In the DeeP-C study, mt-sDNA identified 42.4% of patients with sessile serrated adenomas ≥ 1 cm, compared to only 5.1% detected by FIT (*p* < 0.001), due to mt-sDNA’s capability to detect abnormal methylated DNA in the stool from non-bleeding lesions [71] (Figure 3).

mt-sDNA testing proves to be more sensitive and specific than FOBT for early detection of CRC and advanced adenomas. In comparison with FOBT, Barnell et al. showed higher sensitivity for detecting CRC (90.0% vs. 42.0%) and advanced adenomas (70.6% vs. 19.6%) for mt-sDNA. Additionally, it demonstrated superior specificity in CRC detection (94.0% vs. 90.0%) [72]. No direct adverse effects or complications related to mt-sDNA beyond those associated with follow-up colonoscopy after a positive test were reported. However, the high cost of these tests is a significant limitation for their implementation in screening programs. The price of fecal DNA tests is approximately 15 times higher than that of FIT, making them economically viable only with substantially higher participation rates [73]. False-positive results arising from DNA markers are also problematic, as these markers are not entirely specific and individuals with a positive test will have no explanatory findings on colonoscopy [74]. Hoffman et al. conducted a modeling study, using various equal participation rates based on observational data, and consistently found that screening with FIT or colonoscopy was more effective and less costly than fecal DNA screening, even with a 3-year testing interval [75] (Table 1).

mt-sDNA testing has emerged as a promising noninvasive option for CRC screening due to its high specificity. This technology is increasingly recognized in current screening guidelines. USPSTF has integrated mt-sDNA into its CRC screening recommendations as a novel approach [76,77,78]. The American Cancer Society also endorses mt-sDNA, recommending its use at three-year intervals [40]. In this regard, the studies conducted by Imperiale et al. demonstrate that repeating mt-sDNA screening every three years significantly improves the detection of advanced precancerous lesions [79]. Furthermore, the US Joint Guidelines for CRC and Adenomatous Polyp Screening emphasize that screening strategies should not only detect early-stage cancer but also identify adenomatous polyps [80,81]. The inclusion of mt-sDNA in these guidelines underscores its potential to complement traditional screening methods. As such, mt-sDNA serves as a complementary option to colonoscopy, rather than a replacement. It improves screening rates, especially for average-risk patients who face barriers to undergoing colonoscopy. While mt-sDNA is not intended to replace colonoscopy, it offers a valuable alternative that can enhance screening rates due to its high specificity. This procedure can reduce unnecessary diagnostic procedures and its results are particularly important for average-risk patients who may face barriers to undergoing colonoscopy [82]. By presenting mt-sDNA as an option, clinicians can improve colorectal cancer screening and detection rates, ultimately aiming to reduce the associated morbidity and mortality.

### 2.3. Use and Abuse of Fecal Occult Blood Test

According to guidelines, the FOBT has been approved for screening patients (aged 45 and older) who have an average risk of being diagnosed with CRC [83]. However, FOBT continues to be used for purposes other than CRC screening. With its documented effectiveness as a screening tool, ability to reduce CRC-related mortality, low cost, and non-invasive nature, FOBT has gained widespread use. Unfortunately, it has a potential misuse. The inappropriate use of fecal occult blood testing in patients who are not suitable candidates may lead to additional unnecessary endoscopic investigations with low diagnostic yield and high costs [84]. In this regard, it is important to note that the use of FOBT is validated only for asymptomatic outpatient patients [85]. Its use in symptomatic, hospitalized, and high-risk patients is not recommended as it is not sufficiently sensitive. FOBT is primarily used in the evaluation of possible gastrointestinal bleeding in patients who do not meet the criteria for CRC screening as an additional diagnostic modality in other clinical scenarios [86,87]. Among the diagnoses for patients with positive FOBT results, common colorectal conditions such as hemorrhoids, ulcerative colitis, diverticulitis, and polyps were found, as well as normal conditions [46].

The inappropriate use of FOBT may also delay appropriate patient care due to significant limitations of the test and the possibility of false positive and false negative results [88]. Narula et al. found that 66% of the patients were taking one or more medications that could cause false positive results on the gFOBT test, and that only 2% were following appropriate dietary restrictions, leading to 64% positive results on one or both tests, with endoscopy normal posterior in 11% of cases suggesting false positives [89]. Furthermore, the test showed low sensitivity in detecting true cases of gastrointestinal bleeding, especially in the case of iron deficiency anemia (IDA): 42% of patients with identifiable causes of IDA had false negative results on the FOBT, which highlights the presence of false negatives and the limited usefulness of the test in patients with a high pretest probability of gastrointestinal bleeding. This highlights the need for careful interpretation of FOBT results and its limited effectiveness as a diagnostic tool in certain clinical contexts.

Recently, advanced adenoma (AA) has been recognized as a crucial target for CRC screening. However, FOBT, widely used as a primary screening method due to its non-invasive nature, shows limited sensitivity for detecting AA [90,91]. The loss of fecal hemoglobin from adenomas is significantly associated with their size, number, and advanced characteristics. The sensitivity and specificity for AA are determined by the chosen test threshold and the number of samples collected; these factors determine the number of colonoscopies needed to evaluate positive results [92]. The study performed by Rozen et al. identified adenomas in 294 out of 1204 cancer-free patients, with 99 of them presenting AAs and polyps [93]. It was also observed that patients with adenomas had elevated levels of fecal hemoglobin, which significantly increased with advanced histology, size, pedunculated shape, and multiplicity of the adenomas (*p* < 0.001 for all). With a threshold of 50 ngHb/mL, the sensitivity and specificity for AAs and polyps were 54.5% (95% CI 44.7, 64.7) and 88.1% (95% CI 86.2, 90.1), respectively, when conducting three tests. At higher thresholds, sensitivity decreased, although it significantly increased with the collection of more samples. Conversely, specificity increased at higher thresholds but decreased with the collection of more samples [93]. Additionally, Cao et al. indicate that a large size, left-sided location, and pedunculated morphology of AAs independently contribute to a decrease in the false-negative rate of FOBT [60]. These findings underscore the importance of optimizing test thresholds and sample collection strategies to improve the accuracy of CRC screening and AA detection. Furthermore, Jodal et al.’s studies showed that the polyp detection rate and adenoma detection rate in follow-up examinations were approximately 20 to 37% and 19 to 25%, respectively, and that only 0.25% of adenomas would progress to CRC [33]. The proportions of non-CRC and CRC cases in men were higher than in women, showing a correlation between gender and diagnostic outcomes [94]. Related studies also indicated that the incidence of polyps, adenomas, and CRC is higher in men than in women [95,96]. Therefore, periodic CRC screening and active follow-up examinations could provide effective diagnostic verification, allowing patients to undergo treatment to manage the disease and prevent its progression.

## 3. Discussion

A comprehensive analysis of various fecal occult blood tests (gFOBT, FIT, and mts-DNA) reveals their fundamental utility in the screening and early diagnosis of CRC. These tests provide a non-invasive and accessible tool for the early detection of malignant and premalignant lesions in the gastrointestinal tract, thereby contributing to a reduction in CRC-associated mortality. The regular implementation of screening programs has proven effective in identifying cases at early stages, facilitating timely therapeutic interventions, and significantly improving patient survival rates [97,98]. Different healthcare systems have adopted various approaches to CRC screening, reflecting their unique healthcare structures and population needs [23,37]. For instance, in the USA, options for screening include annual FIT, annual gFOBT, and mt-sDNA tests every 3 years [78]. In contrast, in many EU countries, FIT is often the first line of screening and is typically recommended every 1 to 2 years for individuals aged 50 to 74. If a FIT result is positive, a follow-up colonoscopy is then performed [29,99].

In these terms, the two-steps EU strategy offers a more scalable and less resource-intensive approach, albeit with a reliance on follow-up colonoscopies for positive cases. However, this may result in an increased demand for confirmatory colonoscopies and greater resource expenditure if the FOBT is administered to patients who do not meet the appropriate criteria. It is crucial to recognize that FOBT is a screening test for secondary prevention and is not intended to serve as a diagnostic tool [100,101]. Moreover, its expanding use in hospitalized patients and/or those unsuitable for other gastrointestinal pathologies poses a risk of incorrect result interpretation, potentially leading to misdiagnosis and inappropriate treatment.

The choice of screening method should balance sensitivity, specificity, cost, and patient adherence to optimize CRC screening effectiveness [14,102]. It is essential to consider the differences among various fecal occult blood tests, recognizing the specific advantages and limitations that each test presents. Enzymatic FOBT measures the peroxidase activity of hemoglobin from any source. This makes it susceptible to detecting bleeding from both the upper gastrointestinal tract and the colorectal region. Additionally, the intake of certain foods (red meats, fruits, and vegetables) and medications (non-steroidal anti-inflammatory drugs) can lead to false-positive results [103]. On the other hand, immunological FOBT uses antibodies that specifically detect human hemoglobin, thus avoiding interference from plant peroxidase in the diet. An important limitation of the FOBT is its relatively low sensitivity for detecting early-stage lesions. It is reported that the low sensitivity of FOBT for detecting colorectal neoplasms is approximately 10% for adenomas and 40% to 85% for CRC [41]. In the event of a positive result, a follow-up colonoscopy is necessary, following the two-step strategy [57].

The gFOBT, though widely used and cost-effective, has a lower sensitivity for detecting advanced adenomas and is susceptible to dietary and medication-based interferences, leading to higher rates of false positives and false negatives. FIT offers higher sensitivity (79–93%) and specificity (94–96%) for detecting human hemoglobin and does not require patient preparation, making it more convenient and resulting in higher patient adherence [62]. However, it still requires repetition for effective screening. While FIT has fewer false positives compared to gFOBT, they can still occur, necessitating careful interpretation and follow-up. In addition, FIT may be more effective in detecting CRC and advanced adenomas: sensitivity for detecting advanced adenomas using gFOBT is significantly lower, ranging from 7% to 20%, though specificity is comparable (87% to 98%) [41].

However, the detection rates for advanced adenomas and right-sided serrated polyps with both FIT and gFOBT remain suboptimal. Despite FIT’s higher sensitivity compared to gFOBT, both tests struggle to effectively identify these lesions [104,105]. Advanced adenomas and right-sided serrated polyps often bleed intermittently or minimally, which can result in false-negative results due to insufficient fecal hemoglobin concentrations. This limitation underscores the need for improved screening methods and highlights the importance of combining these tests with other diagnostic approaches to enhance the early detection and prevention of colorectal cancer [60].

The mt-sDNA can detect DNA mutations and epigenetic changes in addition to hemoglobin. Among its key advantages is its high sensitivity for detecting colorectal cancer (up to 92%) and AAs, making it a powerful tool for early detection in CRC [72]. The mt-sDNA test is non-invasive and does not require dietary restrictions or medication adjustments, which can enhance patient compliance. Additionally, its ability to detect DNA markers and hemoglobin increases the likelihood of identifying neoplastic changes that might be missed by other screening methods. It has also proved to be effective for both proximal and distal colon lesions. However, the mt-sDNA test also has notable drawbacks. Its high cost compared to other stool-based tests like FIT and gFOBT can be a barrier to widespread use. The test also has a higher false-positive rate, which can lead to unnecessary follow-up colonoscopies and increased healthcare costs. Moreover, while mt-sDNA has high sensitivity for cancer, its specificity is lower, meaning it may result in more false positives [106,107,108].

These limitations are significant, as false positives can lead to unnecessary additional endoscopic investigations with low diagnostic yield and high costs, while false negatives can delay appropriate care. It underscores the need for continuous improvement and innovation in CRC screening techniques. An ideal CRC screening strategy would integrate the strengths of different tests while addressing their limitations. One approach could involve combining the high sensitivity and patient convenience of FIT with periodic mt-sDNA testing. It may capture cases that might evade detection with FIT alone [109]. This combined approach has the potential to enhance overall detection rates of colorectal cancer and advanced adenomas. Additionally, improving patient education on the significance of adhering to screening protocols, especially regarding the necessity of follow-up colonoscopies after positive non-invasive test results, is crucial. Furthermore, investing in technological advancements to develop more sensitive and specific non-invasive tests would be essential. These innovations could potentially improve the early detection of advanced adenomas and right-sided serrated polyps, thereby reducing the incidence and mortality of CRC and improving patient outcomes (Table 2).

Strategies for CRC screening stand to benefit significantly from the integration of new biomarkers, such as mt-sDNA tests. These advanced biomarkers offer enhanced sensitivity and specificity, potentially identifying CRC and advanced precancerous lesions with greater accuracy than traditional FIT or gFOBT methods. The mt-sDNA test, for instance, detects a combination of genetic mutations, methylation patterns, and hemoglobin levels in stool samples, providing a comprehensive assessment that can improve early detection rates [82]. By incorporating such biomarkers into screening protocols, we can address the limitations of existing tests, such as lower sensitivity for detecting early-stage disease and precancerous lesions [110]. Additionally, these non-invasive tests can increase patient compliance by offering a convenient alternative to more invasive procedures [111]. As a result, strategies based on new biomarkers have the potential to optimize CRC screening, reduce diagnostic delays, and ultimately enhance patient outcomes through earlier and more precise detection of colorectal neoplasms. As research and validation of such biomarkers continue to evolve, they are likely to play an increasingly important role in personalized and effective CRC screening strategies.

Currently, the concept of biomarkers for CRC is constantly evolving and flexible. These biomarkers should be cost-effective and easily applicable for early cancer detection, as well as for determining disease prognosis and predicting treatment response, thereby improving patients’ quality of life [112,113]. During the early stages of colorectal carcinogenesis, epigenetic modifications are more prevalent than genetic mutations, suggesting they could be crucial as new diagnostic biomarkers to identify polyps and colorectal malignancies. In addition to their utility in early detection, biomarkers play a critical role in assessing CRC recurrence, as up to 50% of patients experience recurrence or metastasis, and a quarter of diagnoses occur at advanced stages with unfavorable prognoses and low five-year survival rates [33]. In this context, technological advancements have led to the discovery of various biomarkers that could enhance or supplement early CRC diagnosis, offering potential for more personalized treatments and improved survival rates. Despite these advances, colonoscopy remains the gold standard for CRC diagnosis. Until these biomarkers undergo large-scale validation, colonoscopy will continue to be essential for confirming diagnoses. Nonetheless, the goal is to develop new, efficient, and safe fecal biomarkers for population screening, particularly for younger individuals, where CRC rates have been rising, to facilitate early detection [114].

## 4. Conclusions

Fecal occult blood tests, including guaiac gFOBT, FIT, and mt-sDNA, are essential tools for CRC screening, offering non-invasive methods for early disease detection and mortality reduction. Each technique has its strengths and limitations: gFOBT, while commonly used, has lower sensitivity for advanced adenomas and is affected by dietary and pharmacological factors; FIT offers enhanced sensitivity and specificity with fewer dietary restrictions, improving screening adherence; and mt-sDNA, though highly sensitive for detecting CRC and advanced adenomas, faces limitations due to its higher cost and need for greater patient involvement. To optimize CRC screening, it is important to leverage the strengths of each test while addressing their limitations. Enhancing patient education and adherence, ensuring appropriate use of testing to avoid unnecessary costs and delays, and advancing technology to develop more precise non-invasive tests are crucial steps. These improvements aim to increase early detection rates and ultimately enhance patient outcomes and public health.

## Figures and Tables

**Figure 1 healthcare-12-01645-f001:**
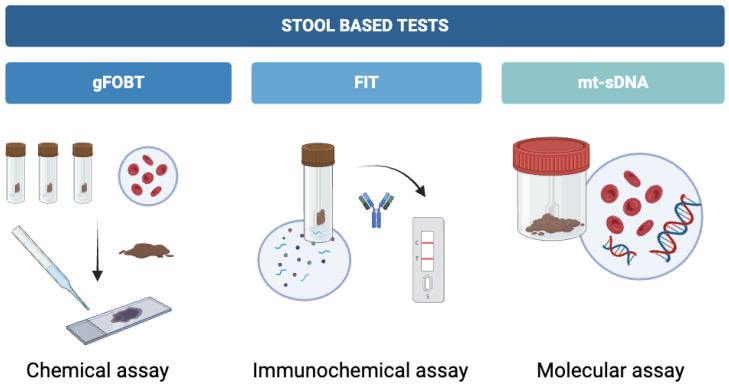
Different types of stool-based tests and the methodologies employed by each, including gFOBT, FIT, and mt-sDNA. gFOBT: guaiac-based fecal occult blood test; FIT: fecal immunochemical test; mt-sDNA: multitarget stool DNA test.

**Figure 2 healthcare-12-01645-f002:**
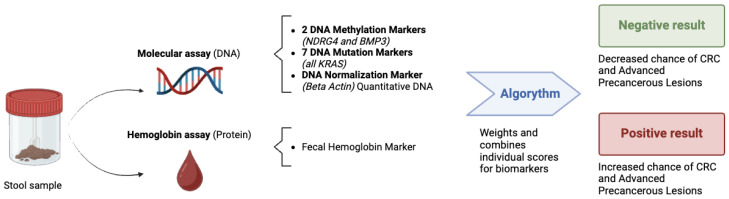
Methodology of mt-sDNA testing: molecular analysis for DNA alterations (methylations and mutations) and immunological analysis for detecting fecal hemoglobin. The results are processed using an algorithm that provides a positive or negative result based on the value relative to a specific cut-off.

**Figure 3 healthcare-12-01645-f003:**
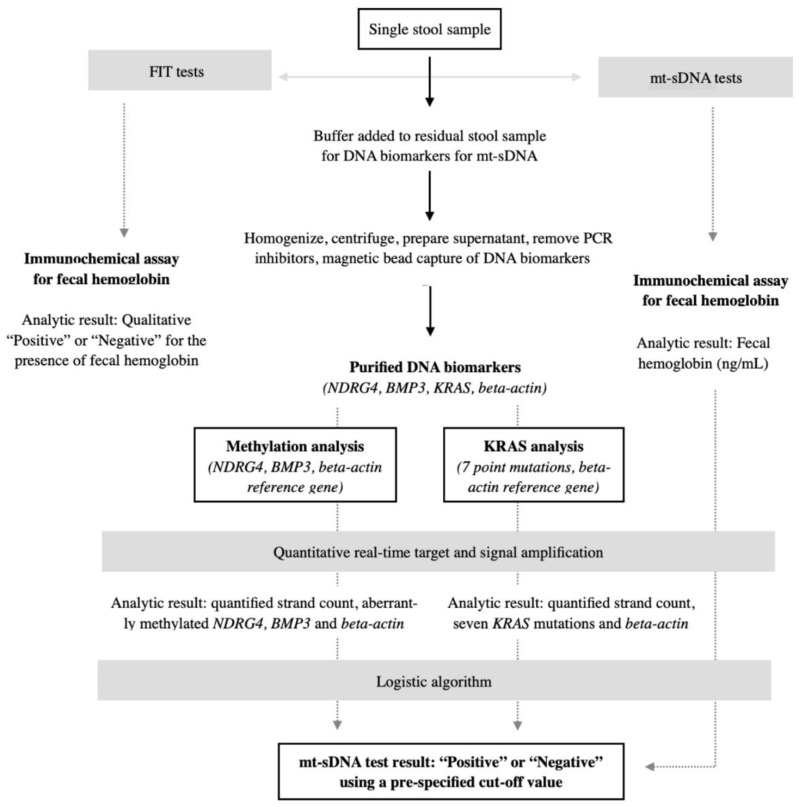
Diagram representing the stool sample flow and approach to extraction and analysis of mt-sDNA and fecal hemoglobin immunoassays. Adapted from Imperiale et al. [68].

**Table 1 healthcare-12-01645-t001:** Sensitivity and specificity comparison for different stool-based screening tests for detection of colorectal cancer. CI: confidence interval. gFOBT: guaiac-based fecal occult blood test; FIT: fecal immunochemical test; mt-sDNA: mt multiTable 73.

	gFOBT	FIT	mt-sDNA
Sensitivity	0.31–0.79 (95% CI, 0.09–1.0)	0.79 (95% CI, 0.69–0.86)	0.93 (95% CI, 0.87–1.0)
Specificity	0.87–0.98 (95% CI, 0.95–0.99)	0.94 (95% CI, 0.92–0.95)	0.84 (95% CI, 0.84–0.86)

**Table 2 healthcare-12-01645-t002:** Comparison of the main characteristics of gFOBT, FIT, and mt-sDNA for colorectal cancer screening. gFOBT: guaiac-based fecal occult blood test; FIT: fecal immunochemical test; mt-sDNA: multitarget stool DNA. Adapted from Studies of colorectal cancer screening, IARC [41].

gFOBT	FIT	mt-sDNA
Chemical reaction	Immunochemical reaction	Molecular assays for KRAS gene mutations and aberrant methylation of NDRG4, SDC2, and TFPI2
Nonspecific for human hemoglobin	Specific for human hemoglobin	Altered DNA and hemoglobin
Qualitative	Qualitative or quantitative	Qualitative
Nonspecific for lower intestinal tract	Higher analytical specificity for lower intestinal tract	Lower specificity and high cost
Lower sensitivity and detection rate for advanced neoplasia	Higher sensitivity for distal lesions and detection rate for advanced neoplasia	The highest sensitivity for proximal and distal colon and detection rate for advanced neoplasia
Avoid consuming red meat for 3 days prior, refrain from high doses of vitamin C, and consider discontinuing NSAIDs or aspirin treatment	Dietary restrictions or medication modification are not required	No dietary nor pharmacological restrictions
Larger stool specimen and less population adherence	Single sample collection and greater population adherence	Single sample single collection at home
Higher rate of false positives and consequent colonoscopy	Lower rate of false positives	Free from false positives solely due to hemoglobin (p.e. intermittent bleeding)

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
