# Peer review of "Utility of Stool-Based Tests for Colorectal Cancer Detection: A Comprehensive Review"

_healthcare, 2024, doi:10.3390/healthcare12161645_

Round 1

Reviewer 1 Report

Comments and Suggestions for Authors

he manuscript, titled "Utility of Fecal Occult Blood Testing: Review of Methods and Limitations for Colorectal Cancer Detection", provides a comprehensive review of the various methods used for screening for colorectal cancer (CRC), with a particular focus on fecal occult blood testing (FOBT). The review examines the principles, efficacy, and limitations of FOBT, as well as other stool-based tests, and their use in clinical settings. The article is well-organized, covering the epidemiology of CRC, various screening strategies, and in-depth discussions on stool-based testing methods.

 Areas for Improvement

1. The manuscript provides a brief overview of other emerging biomarkers and DNA-based tests for colorectal cancer (CRC) detection. However, a more comprehensive analysis of these novel technologies and their potential impact on CRC screening is required.

2. Including more tables and figures could enhance the readability of the manuscript and provide readers with quick reference points. These could include summaries of key points such as the sensitivity and specificity of various tests.

3. In light of the rapid advancements in colorectal cancer (CRC) screening technologies, it would be beneficial to include a discussion of the most recent developments in this area, including any new guidelines or recommendations.

Overall, this manuscript presents a valuable and comprehensive overview of fecal occult blood testing and its significance in colorectal cancer (CRC) screening. With some enhancements in organization, in-depth analysis of newer technologies, and inclusion of additional visual aids, the manuscript could serve as an even more valuable resource for clinicians and researchers working in the field of oncology. The emphasis on evidence-based analyses and critical evaluation of current practices is commendable, as it significantly contributes to the existing body of literature on CRC screening.

Author Response

Comments 1: The manuscript provides a brief overview of other emerging biomarkers and DNAbased
tests for colorectal cancer (CRC) detection. However, a more comprehensive analysis of
these novel technologies and their potential impact on CRC screening is required.

Response 1: Thank you for your insightful comments. We fully agree that a more comprehensive
analysis of emerging biomarkers and DNA-based tests for colorectal cancer (CRC) detection is
crucial for providing a thorough overview of their potential impact on CRC screening.

We have addressed this by expanding the section on novel technologies. Specifically, we have added
detailed information in Section 2.2.3, which now includes a deeper analysis of multitarget stool DNA
(mt-sDNA) technology. Additionally, we have incorporated this information into the Conclusions
section to emphasize the potential of mt-sDNA in improving CRC diagnosis. These additions are
supported by recent literature, including:

1. Dickinson BT, Kisiel J, Ahlquist DA, Grady WM. Molecular markers for colorectal
cancer screening. Gut. 2015 Sep;64(9):1485-94. doi: 10.1136/gutjnl-2014-308075. Epub
2015 May 20. PMID: 25994221; PMCID: PMC4765995.
2. Berger BM, Ahlquist DA. Stool DNA screening for colorectal neoplasia: biological and
technical basis for high detection rates. Pathology. 2012 Feb;44(2):80-8. doi: 10.1097/
PAT.0b013e3283502fdf. PMID: 22198259.
3. Imperiale TF, Ransohoff DF, Itzkowitz SH, Levin TR, Lavin P, Lidgard GP, Ahlquist
DA, Berger BM. Multitarget stool DNA testing for colorectal-cancer screening. N Engl J
Med. 2014;370:1287–1297.
4. Berger BM, Levin B, Hilsden RJ. Multitarget stool DNA for colorectal cancer screening:
A review and commentary on the United States Preventive Services Draft Guidelines.
World J Gastrointest Oncol. 2016 May 15;8(5):450-8. doi: 10.4251/wjgo.v8.i5.450.
PMID: 27190584; PMCID: PMC4865712.
5. Órdenes P, Carril Pardo C, Elizondo-Vega R, Oyarce K. Current Research on Molecular
Biomarkers for Colorectal Cancer in Stool Samples. Biology. 2024;13(1):15. https://
doi.org/10.3390/biology13010015.
6. Mollman BJ. Colorectal cancer screening: The role of MT-sDNA testing. JAAPA. 2023
Aug;36(8):15-20. doi: 10.1097/01.JAA.0000944596.08257.61.
7. Bosch LJW, Melotte V, Mongera S, et al. Multitarget Stool DNA Test Performance in an
Average-Risk Colorectal Cancer Screening Population. Am J Gastroenterol. 2019
Dec;114(12):1909-1918. doi: 10.14309/ajg.0000000000000445.
8. Johnson DH, Kisiel JB, Burger KN, et al. Multitarget stool DNA test: clinical
performance and impact on yield and quality of colonoscopy for colorectal cancer
screening. Gastrointest Endosc. 2017 Mar;85(3):657-665.e1. doi: 10.1016/
j.gie.2016.11.012. Epub 2016 Nov 21. PMID: 278845181.

We believe these enhancements will significantly improve the manuscript by providing a deeper
understanding of these innovative approaches and their implications for CRC screening. Thank you
once again for your valuable feedback.

Comments 2: Including more tables and figures could enhance the readability of the manuscript and
provide readers with quick reference points. These could include summaries of key points such as the
sensitivity and specificity of various tests.

Response 2: Thank you for your suggestion. We concur that enhancing the visual representation of
data can greatly aid in the comprehension of complex information, such as the sensitivity and
specificity of various stool-based tests. In response to your feedback, we have implemented the
addition of the following figures aimed at enhancing the clarity and usefulness of the manuscript:

- Figure 1. This figure provides a comprehensive summary of the techniques discussed in the review.
It visually details different types of stool-based tests, including gFOBT, FIT, and mt-sDNA, along
with the methodologies employed by each. This visual summary allows readers to quickly compare
and contrast the tests, highlighting their distinct characteristics and applications.
- Figure 2. This image illustrates the methodology of mt-sDNA testing. It includes details on
molecular analysis for DNA alterations, such as methylations and mutations, as well as
immunological analysis for detecting fecal hemoglobin. This graphical representation aids in
understanding the complex processes involved in mt-sDNA testing and emphasizes the unique
aspects of this technology.
- Figure 3. A schematic representation of the stool sample flow is provided, depicting the approach
to the extraction and analysis of mt-sDNA and fecal hemoglobin immunoassays. This figure helps
clarify the procedural steps involved in testing, making it easier for readers to follow and
comprehend the workflow and technical processes.

We have also added a new column to Table 2 that addresses aspects related to mt-sDNA technology.
This addition provides a more detailed comparison and analysis of mt-sDNA in relation to other
stool-based tests, highlighting its specific features and performance metrics. This modification helps
to consolidate information and present it in a more structured manner, facilitating a clearer
understanding of how mt-sDNA compares with other technologies.

Additionally, we have included definitions for key terms used to describe test reliability, such as
"specificity," "sensitivity," and "false/true positive rate," in the second paragraph of Section 2.2, titled
"Stool-Based Tests," on page 4. Defining these terms helps to ensure that readers fully understand
these concepts and their relevance to interpreting the results of stool-based tests.

We believe these enhancements address your suggestions effectively and contribute to a more
comprehensive and accessible presentation of the information.

Comments 3: In light of the rapid advancements in colorectal cancer (CRC) screening technologies,
it would be beneficial to include a discussion of the most recent developments in this area, including
any new guidelines or recommendations.

Response 3: Thank you for pointing this out. We agree with this comment. We agree that
incorporating recent advancements and new guidelines in colorectal cancer (CRC) screening
technologies is essential given the rapid progress in this field. This enhancement is important as it
ensures the manuscript reflects the most current practices, guidelines, and innovations, thereby
increasing its relevance and impact for readers.

In response to your suggestion, we have significantly expanded the discussion of mt-sDNA
technology in Section 2.2.3 of the manuscript. This section now provides a thorough analysis based
on the latest sources and guidelines, including:

1. Bibbins-Domingo K, Grossman DC, Curry SJ, et al. Screening for colorectal cancer: US
Preventive Services Task Force Recommendation Statement. JAMA. 2016;315:2564–75.
2. Zauber A, Knudsen A, Rutter CM, Lansdorp-Vogelaar I, Kuntz KM, Writing Committee of the
Cancer Intervention and Surveillance Modeling Network (CISNET) Colorectal Cancer Working
Group. Evaluating the Benefits and Harms of Colorectal Cancer Screening Strategies: A
Collaborative Modeling Approach. Available from: http://www.uspreventiveservicestaskforce.org/
Home/GetFile/1/16450/cisnet-draft-modeling-report/pdf.
3. U.S. Preventive Services Task Force. Topic Update in Progress, Colorectal Cancer: Screening.
Available from: http://www.uspreventiveservicestaskforce.org/Page/Document/draftrecommendation-
statement38/colorectal-cancer-screening.
4. American Cancer Society. Guidelines for the Early Detection of Cancer. Available from: http://
www.cancer.org/healthy/findcancerearly/cancerscreeningguidelines/american-cancer-societyguidelines-for-the-early-detection-of-cancer.
5. Imperiale TF, Lavin PT, Marti TN, Jakubowski D, Itzkowitz SH, May FP, Limburg PJ, Sweetser S,
Daghestani A, Berger BM. Three-Year Interval for the Multi-Target Stool DNA Test for Colorectal
Cancer Screening: A Longitudinal Study. Cancer Prev Res (Phila). 2023 Feb 6;16(2):89-97. doi:
10.1158/1940-6207.CAPR-22-0238. PMID: 36205504; PMCID: PMC9900315.
6. Levin B, Lieberman DA, McFarland B, et al. Screening and surveillance for the early detection of
colorectal cancer and adenomatous polyps, 2008: a joint guideline from the American Cancer
Society, the US Multi-Society Task Force on Colorectal Cancer, and the American College of
Radiology. CA Cancer J Clin. 2008;58:130–60.
7. Mollman BJ. Colorectal cancer screening: The role of MT-sDNA testing. JAAPA. 2023
Aug;36(8):15-20. doi: 10.1097/01.JAA.0000944596.08257.61.

Additionally, we have examined the current implementation of mt-sDNA technology within the
context of CRC screening in the concluding section of the Discussion. We consider this expanded
analysis offers a thorough overview of mt-sDNA’s capabilities that will significantly clarify this
technology relevance and limitations in the early detection of this disease. It also highlights the
potential role that emerging technologies and biomarkers may play in the future of CRC detection. By incorporating a detailed examination of mt-sDNA, we believe it will provide valuable information about the most current and effective screening methods. To avoid redundancy and prevent the repetition of information already covered in the discussion, we have also summarized the Conclusions section.

Reviewer 2 Report

Comments and Suggestions for Authors

This review focuses on the stool analysis methods for the detection of colorectal cancer. It is my understanding that all information included in the study is accurate and provides a comprehensive overview of the key issues related to the topic.

I have outlined a few minor issues that came to my attention during the review process.

1. The title is, in fact, misleading. Although the manuscript does discuss FOBT, the authors also present other fecal test types, which should be included in the title. It is of significant importance, as recent reviews on this subject tend to focus on either FOBT or other types, rather than a comprehensive overview of both. It would be advantageous to place greater emphasis on this point.

2. It would be beneficial to define the terminology used to describe the reliability of the test, i.e "specificity," "sensitivity," "false/true positive rate," and others. In my experience, some readers may require a brief reminder of these concepts to fully comprehend the material. I also needed to check the definitions once more in order to fully understand the article. Brief reminder would be really helpful.

3. The article would benefit from the addition of figures and/or a graphical abstract. It would be beneficial to include a figure that elucidates the aforementioned concept or a comparison of the methods discussed in this review. In the section designated for the "authors' contributions," six authors are credited with the "visualization" part, which, it would appear, were not made for this manuscript.

4. Table 2 includes only gFOBT and FIT, while different types could also be included for this comparison. Furthermore, the addition of a column on the left, describing the discussed aspect in the following rows, would enhance the readability of this table.

5. In the abstract, the authors incorrectly refer to the mts-DNA test as a FOBT subtype.

6. The initial paragraph of the introduction appears somewhat misplaced. It should be either removed or relocated to the section where the FOBT concept is introduced or at the end of the introduction. Initiating the introduction with the CRC statistics will be appropriate.

7. The "Conclusion" section seems redundant in light of the content presented in the "Discussion" section. It could be omitted without compromising the integrity of the paper.

Author Response

Comments 1: The title is, in fact, misleading. Although the manuscript does discuss FOBT, the
authors also present other fecal test types, which should be included in the title. It is of
significant importance, as recent reviews on this subject tend to focus on either FOBT or other
types, rather than a comprehensive overview of both. It would be advantageous to place greater
emphasis on this point.

Response 1: Thank you for your valuable suggestion regarding the title of our manuscript. Based
on your feedback, We have changed the title of the manuscript to “Utility of Stool-Based Tests
for Colorectal Cancer Detection: A Comprehensive Review” in order to better reflect the content
discussed within the paper. We believe this revised title better emphasize the comprehensive
nature of our review and accurately includes all fecal test types discussed in the manuscript.

Comments 2: It would be beneficial to define the terminology used to describe the reliability of
the test, i.e "specificity," "sensitivity," "false/true positive rate," and others. In my experience,
some readers may require a brief reminder of these concepts to fully comprehend the material. I
also needed to check the definitions once more in order to fully understand the article. Brief
reminder would be really helpful.

Response 2: Thank you for pointing this out. We agree with this comment. Therefore, we have
added the following paragraph to explain and clarify these terms on page 4, in the second
paragraph of section 2.2 stool-based tests, at line 13: “[In this regard, specificity and sensitivity
are crucial metrics in evaluating the performance of diagnostic tests. Sensitivity refers to the
test's ability to correctly identify those with the disease (true positive rate), while specificity
measures the test's ability to correctly identify those without the disease (true negative rate). A
false positive occurs when the test erroneously suggests disease presence in a healthy individual,
whereas a false negative happens when the test fails to detect the disease in an affected
individual. These values are essential for interpreting test results accurately and understanding
the clinical implications of test findings.]”

Comments 3: The article would benefit from the addition of figures and/or a graphical abstract.
It would be beneficial to include a figure that elucidates the aforementioned concept or a
comparison of the methods discussed in this review. In the section designated for the "authors'
contributions," six authors are credited with the "visualization" part, which, it would appear,
were not made for this manuscript.

Response 3: We agree with the reviewer’s suggestion. We also believe that the addition of
figures would enhance the clarity and comprehensibility of the article. In response, we have
included the following figures:
• Figure 1: A summary of the techniques described in the review, detailing different types
of stool-based tests and the methodologies employed by each, including gFOBT, FIT,
and mt-sDNA.
• Figure 2: The methodology of mt-sDNA testing, featuring molecular analysis for DNA
alterations (methylations and mutations) and immunological analysis for detecting fecal
hemoglobin.
• Figure 3: A schematic representation of the stool sample flow and the approach to
extraction and analysis of mt-sDNA and fecal hemoglobin immunoassays.
We believe these additions will significantly enhance the article by providing visual summaries
and clarifying the concepts discussed. Additionally, we have developed the information related to
mt-sDNA and emerging techniques, as well as their use in CRC screening, to further elucidate
these concepts. These modifications have been made in section 2.2.3 and at the end of the
Discussion.

Comments 4: Table 2 includes only gFOBT and FIT, while different types could also be included
for this comparison. Furthermore, the addition of a column on the left, describing the discussed
aspect in the following rows, would enhance the readability of this table.

Response 4: Agree. We have, accordingly, modified the Table 2 by including a left-hand column
that elaborates on subsequent aspects acrosss the following rows to enhance this point.

Comments 5: In the abstract, the authors incorrectly refer to the mts-DNA test as a FOBT
subtype.

Response 5: Thank you for pointing this out. We have carefully considered your comments and
made the necessary modifications to the abstract. Specifically, we have clarified the inclusion of
both chemical (guaiac) and immunochemical (FIT) as FOBT techniques, apart from other stoolbased
tests such as multitarget stool DNA (mt-sDNA) in the following paragraph: “This
comprehensive review assesses the utility of stool-based tests in CRC screening, including
traditional fecal occult blood tests (FOBT), both chemical (gFOBT) and immunochemical
techniques (FIT), as well as multitarget stool DNA (mt-sDNA) as a novel and promising
biomarker”. We believe these changes address the concerns you raised and enhance the overall
quality of our manuscript.

Comments 6: The initial paragraph of the introduction appears somewhat misplaced. It should
be either removed or relocated to the section where the FOBT concept is introduced or at the end
of the introduction. Initiating the introduction with the CRC statistics will be appropriate.

Response 6: We agree with the reviewer’s suggestion. We appreciate your insights and
suggestions for improving the clarity and accuracy of our work. Accordingly, we have changed
the location of the initial paragraph at the end of the introduction, as well as introduce other stool
tests such as mt-sDNA: “[Additionally, multitarget stool DNA (mt-sDNA) is also being
introduced as a screening technique for CRC due to its high sensitivity and its potential to
improve early detection rates. Adherence is crucial for the success of screening programs…]”

Comments 7: The "Conclusion" section seems redundant in light of the content presented in the
"Discussion" section. It could be omitted without compromising the integrity of the paper.

Response 7: Thank you for your suggestion. We understand your point and appreciate your
insight. We believe it is important to include a "Conclusion" section to highlight the key points
addressed in the review. Therefore, we have succinctly summarized the main findings to
underscore the points drawn from the overall information examined in the "Discussion" section.
We feel this enhances the clarity of the paper. We hope these modifications are to your
satisfaction and meet your expectations.
